# Molecular Dissection of *TaLTP1* Promoter Reveals Functional *Cis*-Elements Regulating Epidermis-Specific Expression

**DOI:** 10.3390/ijms21072261

**Published:** 2020-03-25

**Authors:** Guiping Wang, Guanghui Yu, Yongchao Hao, Xinxin Cheng, Jinxiao Zhao, Silong Sun, Hongwei Wang

**Affiliations:** State Key Laboratory of Crop Biology, Shandong Key Laboratory of Crop Biology, College of Agronomy, Shandong Agricultural University, Tai’an 271018, China; wgp03165@163.com (G.W.); yuguanghui2009@126.com (G.Y.); hychao96@126.com (Y.H.); chengxx0120@163.com (X.C.); 17863800651@163.com (J.Z.)

**Keywords:** non-specific lipid transfer protein, pavement cell, trichome

## Abstract

Plant epidermis serves important functions in shoot growth, plant defense and lipid metabolism, though mechanisms of related transcriptional regulation are largely unknown. Here, we identified *cis-*elements specific to shoot epidermis expression by dissecting the promoter of *Triticum aestivum* lipid transfer protein 1 (*TaLTP1*). A preliminary promoter deletion analysis revealed that a truncated fragment within 400 bp upstream from the translation start site was sufficient to confer conserved epidermis-specific expression in transgenic *Brachypodium distachyon* and *Arabidopsis thaliana*. Further, deletion or mutation of a GC(N4)GGCC motif at position −380 bp caused a loss of expression in pavement cells. With an electrophoretic mobility shift assay (EMSA) and transgenic reporter assay, we found that a light-responsive CcATC motif at position −268 bp was also involved in regulating pavement cell-specific expression that is evolutionary conserved. Moreover, expression specific to leaf trichome cells was found to be independently regulated by a CCaacAt motif at position −303 bp.

## 1. Introduction

Plant epidermis, representing the boundary between plants and their external environment, regulates the exchange of gas, water and nutrients and serves as a protective barrier [1,2]. Epidermis cells are crucial to key processes in plant development, such as shoot growth, lipid metabolism and cuticle synthesis, as well as defense against hostile biotic and abiotic stresses [1,2,3]. Transcriptomic analyses of *Zea mays* and *A. thaliana* have identified numerous genes involved in the above processes, in which dominant genes have been found to act in plant shoot epidermis rather than in underlying tissues [4,5,6]. Most of a plant’s surface is covered by ground epidermal cells, such as leaf pavement cells or root rhizodermic cells. Further, cell types with specialized structures, such as trichomes and stomatal guard cells on the aerial parts, root hairs and the aleurone layer of seeds, play diverse essential roles in defense, respiration, nutrition and starch degradation, respectively [7]. Over the past few years, studies had been carried out to investigate the differentiation of these specialized epidermal cells, exploring cell-cell communication, cell lineage and the formation of specific transcriptional complexes that trigger specialization [8,9]. In *A. thaliana*, regulatory networks with R2R3 Myb proteins (GL1/WER), bHLH-type transcription factors (GL3/EGL3), homeodomain-leucine zipper transcription factors (GL2) and a WD40 protein (TTG1) have been constructed for trichome and root hair formation [8]. In contrast, though ground epidermal cells cover most of the plant surface, little is known about their differentiation and its transcriptional regulation. Recently, the L1 box, which can be bound by homeodomain-leucine zipper IV transcription factors, has been demonstrated to be involved in regulating embryo epidermis gene expression [10,11]. However, detailed dissection of the *AtML1* promoter suggested that different *cis*-elements, rather than the L1 box, contribute to gene expression regulation at different developmental stages [10]. The *MtML1* promoter from *Medicago truncatula* contributed epidermis-specific regulation in plant shoot, but functional *cis*-element analysis was not reported [12].

Non-specific lipid transfer proteins (nsLTPs) are encoded by a gene family and contain a conserved eight-cysteine motif (8CM) [13,14,15]. These small peptides were originally defined by their capacity to transfer various lipid compounds in vitro and were thought to play pleiotropic functions associated with epidermal cells [16,17]. For over a decade, *nsLTP*s from various species have been reported to be broadly expressed in the epidermis of plant aerial tissue [17,18,19,20,21]. Systematic expression analysis of nsLTP genes from rice and sorghum demonstrated that a majority of the gene family members are type I (9 KDa) nsLTPs with conserved expression in epidermal cells, especially in young growing shoot, indicating a conserved transcription regulatory pathway [22]. The exact physiological functions of nsLTPs remain unclear, though several roles related to plant epidermis functions have been proposed [17]. Knockout of *LTPG1* in *A. thaliana* altered cuticular lipid composition and simultaneously enhanced susceptibility to infection by the fungal pathogen *Alternaria brasssicola*, demonstrating overlapping LTP involvement in cuticle development and defense mechanisms [23]. Nieuwland et al. argued that nsLTPs are involved in a cell-wall loosening mechanism and affect tobacco shoot growth through cell expansion [24]. Other researchers provided evidence that nsLTPs are involved in regulating lipid storage catabolism [25], somatic embryogenesis [26], stigma and pollen adhesion [27], plant signaling [4,28,29] and plant defense against biotic and abiotic stresses [17,24,30].

The multiple functions and relatively conserved expression patterns of nsLTPs indicate that study of expression mechanism might be an important step in understanding the transcriptional regulation of genes participating in plant epidermis processes in aerial tissue. Previously, *nsLTP*s promoter characterization has been performed in transgenic *A. thaliana*, rice and tobacco [19,31]. However, no clearly defined *cis*-regulatory motifs have been identified to date. Previous studies have shown that *TaLTP1* is expressed in epidermis of leaf, shoot and reproductive organs and expression is responsive to diverse biotic and abiotic stresses [15,20]. Characterization and analysis of the *TaLTP1* promoter are required to follow up on these previous studies [15,20]. In this present study, transcriptional regulation of the *TaLTP1* promoter was studied by analyzing mutated and deleted promoters in transgenic *A. thaliana* and *B. distachyon,* as well as with EMSA in vitro. We experimentally defined regions of the *TaLTP1* promoter that mediate tissue-specific expression and identified *cis*-elements responsible for epidermal cell-specific expression in shoot.

## 2. Results

### 2.1. Mapping of a 5’upstream Region Regulating Epidermis-Specific Expression

To identify the proximal part of the promoter, the 1-kb sequence upstream from the *TaLTP1* start codon was searched in the PlantCARE database (http://bioinformatics.psb.ugent.be/webtools/plantcare/html/) and two potential TATA boxes close to the translation start codon were found at −134 and −91 bp upstream. In order to verify the true promoter sequence, we attempted to amplify the full mRNA sequence from a cDNA library using 5’-RACE kit (Takara, Dalian, China). A sequence including the coding region and 60 bp of upstream sequence was amplified, indicating that the TATA box at −91 bp might be the RNA polymerase II binding site (Appendix A).

Previously, we demonstrated that an expression cassette, comprised of the 898 bp upstream of the *TaLTP1* translational start site fused to a GUS reporter gene, drives ubiquitous expression in the epidermal cell layer throughout the transgenic plant life cycle, especially in young growing shoot [20]. To further delimit the upstream regulatory region of *TaLTP1*, a set of seven deletion constructs was generated and transformed into *A. thaliana* for promoter activity analyses (Figure 1a). Both the histochemical GUS analyses and quantitative fluorometric assays showed that the −400*TaLTP1::uidA* construct was able to confer full activity at the vegetative stage (Figure 1b,c) with strong expression in young growing shoot epidermis (Figure 1c1–c3). The −343*TaLTP1::uidA* construct, which further deletes the promoter to −343 bp, caused a loss of quantitative GUS activity (~90%) in transgenic plant leaves that was mainly attributed to the expression in pavement cells (Figure 1c_3_), as trichome expression could still be observed (Figure 1c9). When deleted to position −297 bp (−297 *TaLTP1::uidA* construct), the activity in true leaves disappeared completely, but cotyledon expression was not obviously altered until the promoter region was shortened to position −247 bp (Figure 1c13–c15,c19–c21).

Epidermis-specific expression of *nsLTP*s has been reported for both dicot and monocot plants [12,18,20,32]. Here, we also transformed promoter deletion constructs −400*TaLTP1::uidA,* −343*TaLTP1::uidA*, −297*TaLTP1::uidA*, −247*TaLTP1::uidA* into a monocot, *B. distachyon* (Figure 1c4–c6,c10–c12,c16–c18,c22–c24). The results showed similar promoter activity in both *A. thaliana* and *B. distachyon*, suggesting that conserved functional *cis*-elements regulating pavement cell- and trichome cell-specific expression should be located from −400 bp to −343 bp and −343 bp to −297 bp, respectively. Due to the morphological differences between *A. thaliana* and *B. distachyon* (e.g., there is no cotyledon or petiole in *B. distachyon*), only the conserved expression in the pavement and trichome cells in true leaves was employed to deduce the *cis*-regulation of *TaLTP1* via the promoter mutation assay in *A. thaliana* (Figure 1c).

To further verify that the crucial fragment from −400 bp to −297 bp is sufficient to drive epidermis-specific expression, we fused this promoter fragment to the CaMV 35S mini promoter and transformed this construct into *A. thaliana* to observe promoter activity. However, only cotyledon and trichome expression was observed for this construct, with no activity in pavement cells (Figure 2a). Thus, we then fused a longer promoter fragment of *TaLTP1* (−400 bp to −247 bp) to the CaMV 35S mini promoter for the transgenic reporter assay. The result showed that full epidermis-specific expression, including pavement and trichome cell expression, was conferred by this construct (Figure 2b), indicating that another putative functional *cis*-element is located between −297 bp and −247 bp. Collectively, those evidences indicated two elements that regulate pavement cell-specific expression should be located from −400 bp to −343 bp and from −297 bp to −247 bp and one *cis*-element that regulates trichome-specific expression should be located from −343 bp to −297 bp.

### 2.2. A GC (N_4_) GGCC Motif at −380 bp Is Involved in Regulating Pavement Cell-Specific Expression

To further identify specific *cis*-elements that contribute to epidermis-specific expression, more detailed dissections of the *TaLTP1* promoter were assayed in *A. thaliana* and with electrophoretic mobility shift assays (EMSA) performed using whole-cell plant extracts and biotin-labeled promoter fragments as probe.

Based on the hypothesis that the region from −400 bp to −343 bp harbors functional motifs that regulate pavement cell-specific expression in leaves (Figure 1c), three additional truncated constructs (−380*TaLTP1::uidA*, −372*TaLTP1::uidA* and −359*TaLTP1::uidA*) were transformed into *A. thaliana.* The results showed that deletion to position −372 bp caused a loss of the quantitative expression in leaf pavement cells, demonstrating that the 8 bp fragment between positions −380 bp and −372 bp (or nearby flanking regions) directly determines the activity in pavement cells of young developing leaves (Figure 3a,b). In order to further define the responsible *cis*-element, we created mutations for this small motif (including the above mentioned 8 bp and proximal 2 bp) in the −400*TaLTP1::uidA* promoter construct and monitored reporter activity in transgenic *A. thaliana* (Figure 3c). The results showed that the central nucleotides from −378 bp to −375 bp were less crucial than the flanking sequences for functional regulation of pavement cell-specific expression (Figure 3c). Thus, we defined this motif as GC(N_4_)GGCC.

### 2.3. A CCaacAt Motif at −303 bp Regulates Trichome-Specific Expression

A truncated fragment between positions −343 bp and −297 bp has been shown to be crucial for trace expression in first true leaves, mainly in trichome cells (Figure 1c). Further deletion analysis allowed us to identify that a minimal promoter region from position −305 bp (−305*TaLTP1::uidA*) was sufficient for mediating leaf trichome expression, indicating that the 8 bp fragment between positions −305 bp and −297 bp was the full or at least a partial functional *cis*-element (Appendix A). The putative CA-rich element was deduced as from −303 bp to −297 bp (Appendix A). In order to investigate the potential transcription complex, EMSA was performed by using whole cell protein extracts and a biotin-labeled promoter fragment from −305 bp to −254 bp as probe. We found two bands that could be independently competed out by two complementary unlabeled promoter fragment sequences between positions −305 bp and −254 bp (Figure 4a), implying the presence of two protein binding motifs at two different locations.

Then, a shorter probe was made with −305 bp to −285 bp—containing the CA-rich element and used in EMSA—demonstrating a single shift band (Figure 4b) that represents a putative transcriptional complex at the CA-rich element. To verify this hypothesis, we mutated the flanking sequence of CA-rich element (−296 bp to −287 bp) in the −305*TaLTP1::uidA* promoter construct and found no effect on trichome cell expression. This result suggested that this region is not regulating trichome-specific expression (Appendix A) and delineated this element as −303 bp to −297 bp (Appendix A). Next, we mutated the CA-rich element by replacing it with the *Xho*I restriction enzyme site in the −400*TaLTP1::uidA* and −343*TaLTP1::uidA* promoter constructs, which abolished trichome cell expression but did not apparently affect activity in true leaf pavement cells in transgenic *A. thaliana* harboring the −400*TaLTP1::uidA* construct (Figure 4d_1_,d_3_). These results suggest that the trichome-specific expression regulated by this CA-rich element is independent of pavement cell expression. To define core protein binding sequences in the CA-rich element, we performed single nucleotide substitution analysis via EMSA in vitro with single T/A mutants used as competitors in protein-DNA binding activity tests with the CA-rich element. From comparisons with the band made using probe alone without competitors, the first two C and last A nucleotides in the motif (CCaacAt) were found to be relatively critical for transcription complex formation (Appendix A). Thus, we name this element as CCaacAt motif.

### 2.4. A CcATC Motif at 268 bp Is also Involved in Regulating Pavement Cell-Specific Expression

In addition to the CCaacAt motif, the EMSA result with the promoter fragment from −305 bp to −254 bp as probe indicated the presence of another *cis*-element that forms a transcription complex (Figure 4a,b). We further confirmed this hypothesis by using a short probe (−274 bp to −255 bp) harboring a putative CcATC repeat motif (Figure 4c). A single shift band representing a functional protein-DNA complex was observed for this probe that could be gradually competed out by unlabeled competitor (Figure 4c). Because promoter deletion to position −297 bp (−297*TaLTP1::uidA*) abolished all epidermis-specific activity in true leaves in transgenic *A. thaliana* and *B. distachyon*, we thus attempted to define the functionality of the CcATC motif at position −268 bp by mutating the −400*TaLTP1::uidA* promoter fragment. Briefly, the CcATC motif in construct −400*TaLTP1::uidA* was replaced by the *Xho*I restriction enzyme site and transformed into *A. thaliana* to observe promoter activity. This led to a loss of leaf pavement cell expression (Figure 4d), indicating the CcATC motif, in addition to the GC(N_4_)GGCC motif at position of −380 bp, was involved in regulating pavement cell-specific expression (Figure 3). In contrast, deletion of this motif in −400*TaLTP1::uidA* or −343*TaLTP1::uidA* constructs did not affect trichome expression (Figure 4d_2_,d_4_), suggesting that the GC(N_4_)GGCC motif is not involved in regulating trichome-specific expression.

### 2.5. Characterization of the CcATC Motif

To define core protein binding sequences in the CcATC motif that regulate pavement cell-specific *TaLTP1* gene expression, we analyzed a series of nucleotide substitutions with EMSA in vitro. These results revealed a consensus CcATC motif, with the first C and the A, but not the second C, appearing to be essential for protein binding (Figure 5b). Alignment of *nsLTP* promoters from different species showed that this motif often appears as C(cATC)_2_ with the first C nucleotide in the CATC repeat not being conserved (Figure 5c), consistent with the EMSA results (Figure 5b).

Semi-quantitative RT-PCR had previously shown that *TaLTP1* transcription was induced by light and repressed by dark treatment [20]. Here, *TaLTP* up-regulation by light in native plants was verified by real-time PCR (Appendix A). To determine if light-induced expression changes were due to promoter *cis*-regulation, we applied dark treatments to transgenic *A. thaliana* containing the *TaLTP1* promoter 5’ sequence deletion series and quantified the expression. Our results showed that the *TaLTP1* promoter could confer light inducible expression in transgenic *A. thaliana* (Appendix A). Under dark, the seedlings of transgenic lines harboring 5’ sequence deletion mutants (−898 bp, −725 bp, −512 bp and −400 bp) showed obviously decreased GUS activity in histochemical and quantitative analyses. However, expression differences between light and dark condition were not observed upon further deletion with construct −343*TaLTP1::uidA*, which only drove trichome-specific expression (Appendix A). This result suggests that light regulation of *TaLTP1* is associated with quantitative expression in leaf pavement cells.

Because the CcATC motif had been identified as a regulator of pavement cell-specific expression, it was possible that this *cis*-element also plays a role in regulating epidermis-specific light-inducible expression. To test this in native plants, we investigated protein-DNA binding activities under light conditions. Seedlings treated with 24 or 48 h of constant light were harvested and equivalent amounts of whole cell protein extracts were applied in EMSA. The result showed that the amount of protein-DNA complex formed at the CcATC motif increased under light conditions (Appendix A).

## 3. Discussion

Many nsLTPs genes have been reported to exhibit conserved plant shoot epidermis expression, including those from both dicot and monocot plants [13,17]. This was confirmed by our *TaLTP1* promoter transgenic assays with *B. distachyon* and *A. thaliana* (Figure 1). Quantitative and histochemical GUS analysis of *TaLTP1* promoter deletion constructs showed that pavement cell and trichome expression are independently regulated by different *TaLPT1* promoter *cis*-elements, wherein the quantitative expression in leaves was mainly attributable to that in pavement cells (Figure 1).

Our deletion analysis of the *TaLTP1* promoter showed that the −400 bp upstream fragment was sufficient to mediate expression in leaf epidermis, including distinct pavement cell and trichome expression (Figure 1). Further dissection of this promoter region allowed us to identify a GC(N_4_)GGCC motif at −380 bp that was crucial for regulating pavement cell-specific expression in leaf (Figure 3). Further, both transgenic promoter assays and EMSA showed that this pavement cell-specific expression was not only regulated by the GC(N_4_)GGCC motif but also by an evolutionary conserved and light-responsive CcATC motif at −268 bp (Figure 4 and Figure 5). The regulation from this pair of *cis*-elements, which determined pavement cell expression, is apparently distinct from previously reported L1 box 5’-TAAATG(C/T) regulation of pavement cell-specific expression in maize kernel and *Arabidopsis* embryo [10,11]. In the future, transcription factors binding these motifs should be identified, and potential molecular interactions between these two motifs or transcriptional complexes should be elucidated.

Complete dissection of the *TaLTP1* promoter with transgenic *A. thaliana* and gel retardation assays allowed us to identify another *cis*-element, a CCaacAt motif location at position −303 bp (Figure 4). Our results clearly support that this motif’s regulation of trichome-specific expression is independent of pavement cell regulation (Figure 4 and Appendix A). With single nucleotide substitution and EMSA, the consensus for this motif was defined as CCaacAt. This element overlaps with a previously identified RAV1AAT CAACA consensus motif, which can be bound by RAV transcription factors [34]. However, RAV transcription factor involvement in plant epidermis regulation is largely unknown and potential binding between the CCaacAt motif and RAV transcription factor needs to be further investigated.

## 4. Materials and Methods

### 4.1. Generation of Deletion and Nucleotide Substitution TaLTP1 Promoter Constructs

The *TaLTP1* promoter::*uidA* construct has been described previously [20]. Fragments deleting 5’ regions of the putative *TaLTP1* promoter were amplified by specific primers (Appendix A) and cloned into pGEM-T easy vector (Promega, Madison, WI, USA) or pMD18-T vector (Takara, Dalian, China). Site-directed mutagenesis was done following the QuickChange II site-directed mutagenesis kit protocol (Agilent, La Jolla, CA, USA). For mutations in Figure 3c, primary mutated plasmids were amplified by specific overlapping mutant primer pairs GC-M1-F, GC-M1-R to GC-M5-F, GC-M5-R (shown in Appendix A), using pMD18-T vector harboring *−400TaLTP1* as template. Then, the primary mutated plasmids were digested with *Dpn*I restriction enzyme and transformed into XL1-Blue supercompetent cells (New England Biolabs, Beijing, China). Mutated fragments with the *Hind*III and *Xba*I restriction enzyme were eventually subcloned into the binary vector pBI121, constructing mutant expression vectors M1–M5. For mutant expression vectors (m1–m5) shown in Appendix A, primary mutated plasmids were generated with template pGEM-T easy vector harboring *−305TaLTP1*, amplified by specific primers detailed in Appendix A, then the mutated fragments were also subcloned into the binary vector pBI121 with the *Hind*III and *Xba*I restriction enzyme. In −400*TaLTP1*::*uidA* and −343*TaLTP1*::*uidA* (Figure 4d), CCaacAt and CcATC motifs were replaced by the *Xho*I restriction enzyme site. For the −400mCCaacAt mutation, for instance, two PCR products amplified with LTPp400-F and 400/343mCCaacAt-*Xho*I-r, 400/343mCCaacAt-*Xho*I-f and LTPp-*Xba*I-R were inserted into the binary vector pBI121. PCR amplification was composed of a 3 min denaturing step at 95 °C, 32 cycles of amplification (20 s at 95 °C, 20 s at 56 °C and 30 s to 3 mim at 72 °C) and a 10 min at 72 °C. Finally, the expression vectors pBI121 harboring each fragment were transformed into *A. thaliana* (Col-1) using the *Agrobacterium*-mediated floral dip method [35]. Alternatively, using restriction enzyme sites for *Hind*III and *Bam*HI, promoter fragments were inserted into pCAMBIA 1391Z for further transformation in *B. distachyon*. The −46 bp minimal CaMV 35S promoter was generated by PCR using pBI121 plasmid as template. The PCR product was digested with *Xho*I and *Xba*I and the digested fragment was used to replace the *Xho*I–*Xba*I fragment in pBI121. Truncated fragments from −400 to −147, −196 or −247 bp was subcloned into the *Xho*I site located upstream of −46:35S*:uidA* to generate constructs −147:35S*:uidA,* −196:35S*:uidA* or −247:35S*:uidA*, respectively.

### 4.2. Plant Growth and Transformation

*A. thaliana* (Col-0) was grown at 25/22 °C under a 16-h light/8-h dark cycle until flowering stage. Agrobacterium-mediated transformations were performed by using the flower-dip method as described previously [35]. Seeds of transformants were sterilized and selected on 0.8% (*w*/*v*) half-strength MS agar medium supplemented with 1% (*w*/*v*) sucrose and 50 mg/L kanamycin sulfate. The pBI121 vector harboring the CaMV35S::*uidA* reporter gene and pBI101 vector without the uidA gene were also transformed into *A. thaliana* as positive and negative controls. For each truncated promoter::GUS construct, three independent lines of transgenic *A. thaliana* were selected for quantitative measurement of GUS activity, and at least 10 independent transgenic lines were used for histochemical GUS analysis. Moreover, the TaLTP1 promoter fragments inserted in pCAMBIA 1391Z vector, the bin vector as negative control, were transformed in B. distachyon (Bd21) via Agrobacterium-mediated method [36]. Three to five independent transgenic lines were used to investigate the GUS expression pattern.

### 4.3. Expression Analysis by Real-Time qPCR

Seeds of the ‘Coker797’ wheat cultivar were grown for 10 days with a photoperiod of 14/10 h and a temperature of 25/23 °C (day/night) prior to treatments. For dark condition, the plants were covered by foil and grown in the same chamber for three days. Total RNA was extracted from the leaves using the RNeasy plant mini kit (Qiagen, Hilden, Germany) followed by DNase I digestion. First strand cDNA was synthesized from 1 μg of total RNA (first strand cDNA synthesis kit for RT-PCR, Takara, Dalian, China) and then real-time PCR was performed in a Chromo 4 real-time PCR detection system (Bio-Rad, Waltham, MA, USA) using SYBR Supermix (Bio-Rad, Waltham, MA, USA). PCR amplification of *TaLTP1* and endogenous control was comprised of a 2-min denaturing step at 95 °C, followed by 40 cycles of 10 s at 95 °C and 30 s at 60 °C. The expression level was calculated by the comparative cycle threshold (*C*_t_) method with the wheat 18S rRNA gene as reference.

### 4.4. Histochemical GUS Analyses and Fluorometric Assays

At least 10 independent transgenic lines (T_2_) were immersed in staining buffer (50 mM Na-phosphate, pH 7.2, 3 mM K_3_Fe(CN)_6_, 3 mM K_4_Fe(CN)_6_, 0.5% Triton X-100 and 2 mM X-Gluc) and incubated overnight at 37 °C. Chlorophyll was removed with an increasing ethanol concentration series. The typical GUS staining pattern for most transgenic lines was justified as the expression pattern for the corresponding promoter construct and photographed. For sectioning, the GUS stained tissue samples were embedded in parablast and sectioned with a thickness of 10 μm as described elsewhere [20]. To determine the GUS quantity in plant tissues, harvested seedlings were ground in liquid nitrogen and GUS activity (hydrolysis of 4-methylumbelliferone glucuronide per mg protein per hour) was measured at 460 nm with a Labsystems Fluoroskan II fluorescent microplate reader (Thermo Scientific, Missouri, TX, USA) [37].

### 4.5. Electrophoretic Mobility Shift Assay (EMSA)

All 3’-biotin-labeled probes and unlabeled competitors were commercially synthesized (Sangon Biotech, Shanghai, China). Single-stranded DNA fragments (100 pM) were dissolved in TE buffer and annealed by heating to 94 °C followed by slow cooling to room temperature in a PCR machine. Whole-cell plant extracts were prepared with a Plant Protein Extraction Kit (Pierce, Rockford, IL, USA). All EMSA experiments were performed following LightShift Chemiluminescent EMSA Kit instructions (Pierce, Rockford, IL, USA). Each reaction mixture contained 1× binding buffer, 500 ng of poly(dI-dC), 25 fmol probe and appropriate amounts of glycerol, DNA competitors and protein preparations. The electrophoresis was carried out on 6% polyacrylamide gels in Tris-borate-EDTA buffer (45 mM Tris-borate/1 mM EDTA), which were electro-blotted onto Biodyne B membrane (Thermo Scientific, Walttham, MA, USA). Signal development followed the EMSA kit protocol (Pierce, Rockford, IL, USA) with BioMax films (Kodak, Rochester, NY, USA) used for luminescence detection.

## Figures and Tables

**Figure 1 ijms-21-02261-f001:**
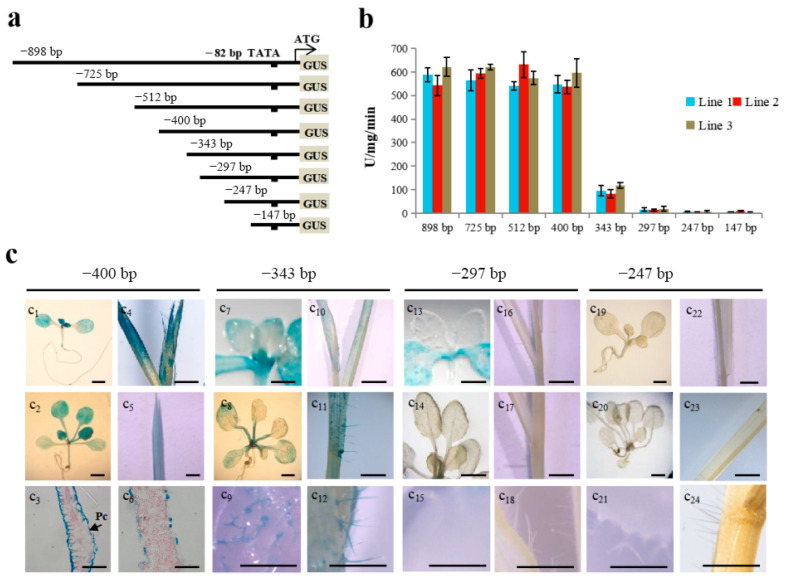
Activities of a *TaLTP1* promoter deletion series in transgenic *A. thaliana* and *B. distachyon*. (**a**) Each of the 5′ upstream fragments from the *TaLTP1* promoter was fused to the translational start site of the beta-glucuronidase reporter gene. (**b**) Quantitative GUS activity of two-week-old T_3_ transgenic *A. thaliana* seedlings with *TaLTP1* promoter deletions. 4-MUG was used as substrate in the assay. Three replicates were used for each transgenic line; each replicate contained at least 5 seedlings. (**c**) GUS activity of *A. thaliana* transformed with pBI121 vector harboring -400*TaLTP1::uidA* (c_1_–c_3_), -343*TaLTP1::uidA* (c_7_–c_9_), -297*TaLTP1::uidA* (c_13_–c_15_), -247*TaLTP1::uidA* (c_19_–c_21_); GUS activity of *B. distachyon* transformed with pCAMBIA 1391Z vector harboring -400*TaLTP1::uidA* (c_4_–c_6_)*,* -343*TaLTP1::uidA* (c_10_–c_12_), -297*TaLTP1::uidA* (c_16_–c_18_), -247*TaLTP1::uidA* (c_22_–c_24_). The first roll represents GUS staining in young seedling leaves, the second roll represents GUS staining in adult leaves of 2- to 3-week-old seedlings, and the third roll represents expression in leaf pavement cells and trichome cells in true leaves. Scale bars are about 40 μm for C_3_ and C_6_, and about 1 mm for all others.

**Figure 2 ijms-21-02261-f002:**
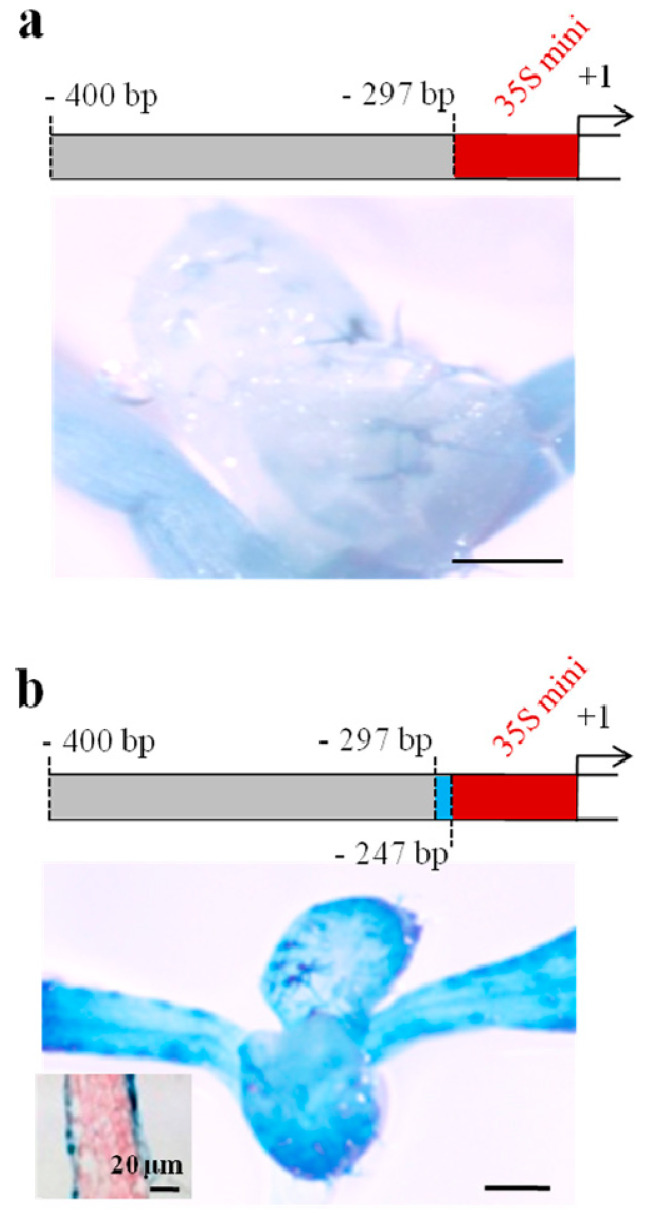
Activities of *TaLTP1* promoter fragment fused to minimal promoter of CaMV 35S in transgenic *A. thaliana.* The promoter fragment from positions −400 bp to −297 (**a**) /−247 bp (**b**) were fused to CaMV 35S minimal promoter and transformed into *A. thaliana* for activity examination. Scale bar = 200 μm.

**Figure 3 ijms-21-02261-f003:**
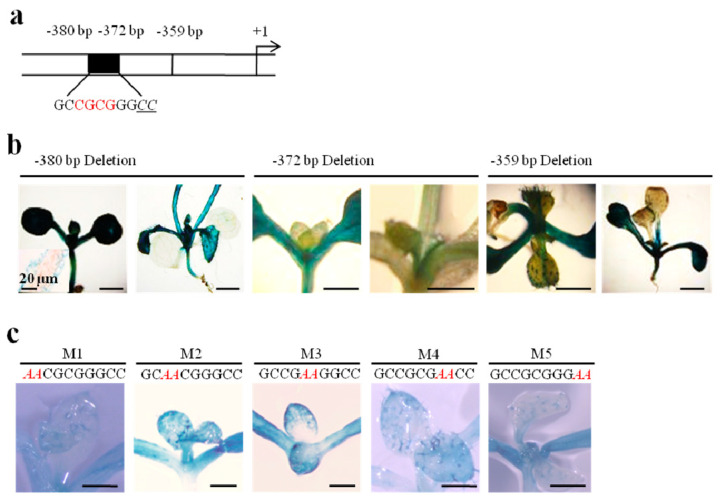
Activities of *TaLTP1* promoter deletion and mutation series in transgenic *A. thaliana.* Identification of a GC(N4)GGCC motif that regulates pavement cell-specific expression. (**a**) Diagram of the GC(N4)GGCC box located within the upstream sequence of *TaLTP1*; (**b**) Histochemical analysis of GUS activity of *TaLTP1* promoter deletion mutants at positions −380 bp, −372 bp and −359 bp. Scale bar = 1 mm; (**c**) GUS activity in *A. thaliana* transformed with five −400*TaLTP1*::*uidA* constructs with mutated GC(N4)GGCC motifs (M1–M5). Scale bar = 200 μm.

**Figure 4 ijms-21-02261-f004:**
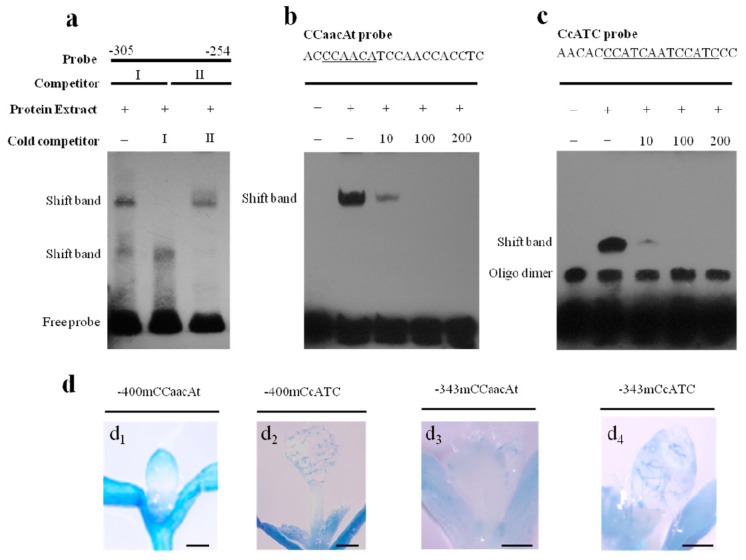
Epidermis-specific expression regulated by CCaacAt and CcATC motifs. (**a**) EMSA using upstream sequence from −305 to −254 bp as probe and non-labeled promoter fragments (−305 to −280 bp, −279 to −254 bp) as competitors. Whole cell proteins were extracted from two-week-old wheat seedlings; (**b**,**c**) EMSA using short probes containing the CCaacAt motif and CcATC motif (−305 to −285 bp, −274 to −254 bp). Unlabeled fragments with the same sequence as probe were used as competitors. A plus sign (+) represents the sample was added and a minus sign (−) represents not; (**d**) GUS activity of *A. thaliana* transformed with −400*TaLTP1*::*uidA* harboring mutated CCaacAt motif (d_1_), CcATC motif (d_2_), and −343*TaLTP1*::*uidA* harboring CCaacAt motif (d_3_), CcATC motif (d_4_). Mutant construct development is described in Materials and Methods. Scale bar = 200 μm.

**Figure 5 ijms-21-02261-f005:**
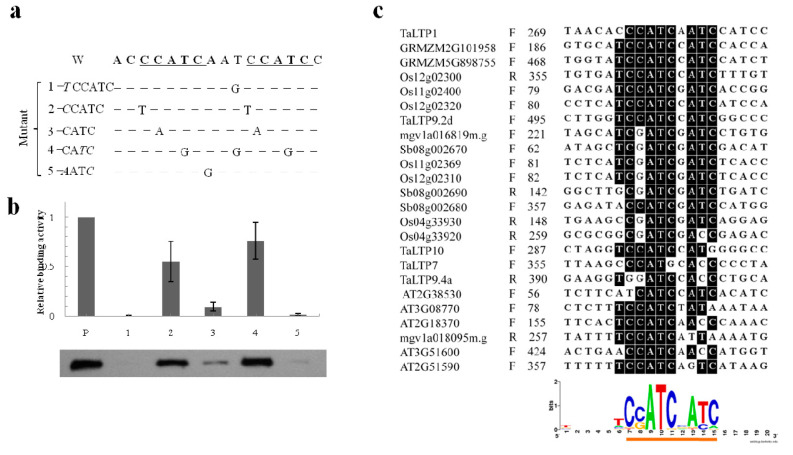
Characterization of the CcATC motif by EMSA using mutated competitors. (**a**) Mutated nucleotides in competitor sequences; (**b**) Relative binding activities in cell extracts when using mutated competitors in EMSA. The band strength was calculated by ImageJ software and was compared with the band formed using probe alone. W, wild type; P, probe. Representative EMSA band patterns when mutated competitors were used are shown below; (**c**) Alignment of CcATC sequences. Conserved nucleotides are shaded. Numbers to the left of the alignment indicate the position of the CcATC with respect to the translation start codon. F, forward strand; R, reverse strand. Sequence logo of the CcATC motif and flanking sequences illustrating conserved nucleotides as taller letters. The CcATC consensus is underlined. The nucleotide sequence logo was created by the WebLogo program (http://weblogo.berkeley.edu/) [33].

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
