# Peer review of "Molecular Dissection of TaLTP1 Promoter Reveals Functional Cis-Elements Regulating Epidermis-Specific Expression"

_ijms, 2020, doi:10.3390/ijms21072261_

Round 1
Reviewer 1 Report
In this manuscript, authors investigated the cis-regulation mode of TaLTP1 promoter by promoter mutation analysis in transgenic Arabidopsis thaliana, Brachypodium distachyon and EMSA in vitro. Where they demonstrated that the regions of TaLTP1 promoter that mediate the tissue specific expression and identified the cis-elements responsible for epidermal cell specific expression in shoot.
The paper, however, must be improved in terms of writing since some grammar and syntax errors are present in the manuscript. Also, a better description of the results should be considered. Furthermore, the discussion section must be more focused on your own data. For instance, explain how all the variables that were measured correlate among each other in light of the observed results? I kindly suggest reorganizing and shortening this section.
Line 13: Delete “et al”.
Line 26: Replace words “epidermis; and cis-element”, because they are already inserted in the title.
Line 29: Delete the expression “(outmost cell layer)”.
Lines 31-33: REFERENCE?
Line 33: replace maize with Zea mays.
Line 64: Correct the reference “Jeroen et al.”[??].
Line 85: Bring figure supplementary 1 to the body of the text.
Although the figures and tables in this manuscript were clear and well-organized, the description of the data in the Results part was under satisfaction. Briefly, most of the description of data is too general and vague to be understood easily or to be accurate. Also, some of the results are plain description/list of numbers and figures, without rationale and conclusion.
Lines 278-290: Describing the PCR conditions, the cloning conditions, the binding conditions ... is very superficial.
Line 300: RT-qPCR.
Lines 301-309: Because it is a relative expression, what method is used? What was the calibrator sample?
Author Response
Point 1:
In this manuscript, authors investigated the cis-regulation mode of TaLTP1 promoter by promoter mutation analysis in transgenic Arabidopsis thaliana, Brachypodium distachyon and EMSA in vitro. Where they demonstrated that the regions of TaLTP1 promoter that mediate the tissue specific expression and identified the cis-elements responsible for epidermal cell specific expression in shoot.
The paper, however, must be improved in terms of writing since some grammar and syntax errors are present in the manuscript. Also, a better description of the results should be considered. Furthermore, the discussion section must be more focused on your own data. For instance, explain how all the variables that were measured correlate among each other in light of the observed results? I kindly suggest reorganizing and shortening this section.
Response 1: We first take this opportunity to thank the reviewer for the encouragement and helpful guidance. We have now revised the manuscript with more clear description of the results. We have now shortened the discussion section as suggested (Line 262-282).
Point 2: Line 13: Delete “et al”.
Response 2: According to the comment, it has been revised in the MS (Line 13).
Point 3: Line 26: Replace words “epidermis; and cis-element”, because they are already inserted in the title.
Response 3: Following the comment, the keywords “epidermis” and “cis-element” have been replaced as “trichome” in the Keywords (Line 24).
Point 4: Line 29: Delete the expression “(outmost cell layer)”.
Response 4: Following the comment, the expression “(outmost cell layer)” has been deleted in the MS (Line 27).
Point 5: Lines 31-33: REFERENCE?
Response 5: Following the comment, the references has been added in the MS (Lines 28, 30).
Point 6: Line 33: replace maize with Zea mays.
Response 6: Following the comment, it has been revised to “Zea mays” in the MS (Line 31).
Point 7: Line 64: Correct the reference “Jeroen et al.”[??].
Response 7: Following the comment, the reference has been added in the MS (Line 63).
Point 8: Line 85: Bring figure supplementary 1 to the body of the text.
Response 8: Following the comment, Fig. S1 has been described in the text (Line 88).
Point 9: Describing the PCR conditions, the cloning conditions, the binding conditions ... is very superficial.
Response 9: Following the comment, PCR conditions has been added in the MS (Line 302-304).
Point 10: Line 300: RT-qPCR.
Response 10: Following the comment, it has been described in the text (Line 322).
Point 11: Lines 301-309: Because it is a relative expression, what method is used? What was the calibrator sample?
Response 10: Following the comment, the method and the reference has been described in the text (Line 330-332).
We again thank the reviewer for the careful and helpful guidance to improve our study and manuscript.
Reviewer 2 Report
Dear the editor,
The manuscript entitled “Molecular dissection of TaLTP1 promoter reveals functional cis-elements regulating epidermis-specific expression” describes the activity of TaLTP1 promoter in epidermal cells in Arabidopsis and Brachypodium. Then, by using promoter-deletion and mutation transgenic Arabidopsis, cis-regulatory elements in the promoter were shown. In addition, from the results of EMSA, two additional cis-regulatory elements were identified. The findings in the manuscript seem to be useful for the relevant researchers, but I felt that some provided data were insufficient for evaluation of the findings.
In the manuscript, major results came from promoter-GUS analyses in transgenic Arabidopsis. However, the exact number of transgenic plants tested in each experiments and what kind of variety in the staining patterns observed in independent transgenic plants are not clearly described. In general, independent transgenic plants show somehow different staining patterns and intensity. Without any information of experimental variety, it is difficult to verify the results and conclusions. Please provide exact number of independent transgenic plants tested in each assay. Moreover, please provide any differences in GUS staining patterns among the transgenic plants independently transformed with the same construct.
Next, Figure 1c, Figure 2c and Figure 3d show pictures difficult to be understood.
In particular, Figure 1c needs more explanation to each panel in the figure legend.
Associated with Figure 2c, the authors argued that GC(N4)GGCC motif regulates pavement cell expression. In Figure 2c M1to M5 panels, GUS staining was seen in petioles. Are the stained cells in petioles not pavement cells?
In all four panels of Figure 3d, what tissues are stained? Was GUS staining in young leaf primordia and petioles not in epidermal cells? Please show clearer and more informative pictures in this figure.
Concerning names of cis-element motifs, I found several possible confusable statements.
Please revise the statements to clarify the manuscript.
- In Abstract, “GGGCC motif” was used, but in result section, GC(N4)GGCC was used.
- I am not sure that CA-rich motif is practically useful. I found that in Figure 3b, the CA rich probe has four CA sequences. On the other hand, in the next figure, figure 3c, the CATC probe also has four CA sequences.
- The name of CATC motif needs to be reconsidered because EMSA and evolutionary conserved sequences suggest CcATC may be more important rather than only CATC.
In the discussion section, the last and second last sentences (line 272-) seem novel results. Please provide this information in the result section.
Some minor points,
- Line 81, please provide information how two TATA boxes were identified.
- Line 272, Arabidopsis MYB94 and MYB96 proteins should be written in capital letters.
- In Figure 2 title, remove B. distachyon.
- In figure S1, please explain what is SORB 1.
- In figure S4a, an error bar is missing.
Author Response
Point 1:
The manuscript entitled “Molecular dissection of TaLTP1 promoter reveals functional cis-elements regulating epidermis-specific expression” describes the activity of TaLTP1 promoter in epidermal cells in Arabidopsis and Brachypodium. Then, by using promoter-deletion and mutation transgenic Arabidopsis, cis-regulatory elements in the promoter were shown. In addition, from the results of EMSA, two additional cis-regulatory elements were identified. The findings in the manuscript seem to be useful for the relevant researchers, but I felt that some provided data were insufficient for evaluation of the findings.
In the manuscript, major results came from promoter-GUS analyses in transgenic Arabidopsis. However, the exact number of transgenic plants tested in each experiments and what kind of variety in the staining patterns observed in independent transgenic plants are not clearly described. In general, independent transgenic plants show somehow different staining patterns and intensity. Without any information of experimental variety, it is difficult to verify the results and conclusions. Please provide exact number of independent transgenic plants tested in each assay. Moreover, please provide any differences in GUS staining patterns among the transgenic plants independently transformed with the same construct.
Response 1: We thank the reviewer for the insightful review and helpful comments. Three independent transgenic lines had been used as three replicates for quantitative evaluation of GUS activities in Fig. 1b, in which each replicate contains at least 15 seedlings. To make clarity, we have now displayed the results in respective to three independent transgenic lines for each promoter construct (Fig. 1b).
We conducted the work of TaLTP1 promoter since 2008. For GUS staining pattern, we generally speculated at least 10 seedlings for each promoter construct as the transformation efficency of A. thaliana is really high. The whole transgenic seedlings would be stained for GUS activity observation. As commented, there was potential variance of the GUS staining between different transgenic lines. Even there is certain variance in the quantitative GUS activity, most transgenic lines showed similar expression pattern in tissue-specificity and in histochemistry. In this study, we only focus to analyze the conserved expression in true leaves, and showed the representative photograph in the figures.
Generally, the cis-elements are difficult to be functionally identified due to the small sizes (~ 4-10 bp). Using the current methology, we did obtain logical results. Mutations of the whole elements or certain nucleotides had caused loss of the correlated function (Fig. 3c, Fig. 4d, Fig. 5b), which in turn evidenced the validity of our results. According to the comment, we have described the methods for GUS staining with more details (Line 334, 337).
Point 2: Next, Figure 1c, Figure 2c and Figure 3d show pictures difficult to be understood.
In particular, Figure 1c needs more explanation to each panel in the figure legend.
Response 2: Thanks for the comment. The legends for these figures have now been revised with detail explanations (Line 103-114, Line 160-163, Line 199-201). The method for achieving the promoter mutants were detailly described in Materials and Methods (Line 288-304).
Point 3: Associated with Figure 2c, the authors argued that GC(N4)GGCC motif regulates pavement cell expression. In Figure 2c M1 to M5 panels, GUS staining was seen in petioles. Are the stained cells in petioles not pavement cells?
Response 3: As reported previously, the epidermis specific expression in aerial tissue is conserved in many nsLTPs in plants, including both dicot and monocot. In this study, we employed the Arabidopsis and brachypodium system to investigate the cis-regulation of TaLTP1. Due to different architecture and potential different transcriptional mechanism between dicot and monocot plants, we here only focus on studying the conserved expression in true leaves between Arabidopsis and brachypodium, as revealed in Fig. 1. In Fig. 3c (previous Fig. 2), we found mutations of different nucleotides in the motifs did altered the expression in pavement cells of true leaf (Line 157). We did not verify if the expression in petioles. According to our previous study, this expression feature in epidermis should cover most of the tissues including root tip, leaf, shoot and silique (Genetica, 2010, 138, 843-852).
Point 4: In all four panels of Figure 3d, what tissues are stained? Was GUS staining in young leaf primordia and petioles not in epidermal cells? Please show clearer and more informative pictures in this figure.
Response 4: In Fig. 4d (previous Fig. 3d), the whole transgenic seedlings were stained. The mutations for CCaacAt motif or CcATC motif were created in two promoter deletion constructs -400 (driving expression in pavement cell and trichome in true leaf) and 343 bp (driving expression in trichome in true leaf). For the -400mCCaacAt construct, the transgenic plant showed quantitative expression in pavement cells. However, for -400mCcATC, -343mCCaacAt, -343mCcATC, we did not find obviously expression in pavement cells (Line 193). Here, we only analyzed the GUS expression pattern in true leaves but not in petioles.
Point5: Concerning names of cis-element motifs, I found several possible confusable statements. Please revise the statements to clarify the manuscript.
In Abstract, “GGGCC motif” was used, but in result section, GC(N4)GGCC was used.
Response 5: According to the comment, it has been revised as GC(N4)GGCC in the abstract (Line 18).
Point6: I am not sure that CA-rich motif is practically useful. I found that in Figure 3b, the CA rich probe has four CA sequences. On the other hand, in the next figure, figure 3c, the CATC probe also has four CA sequences.
Response 6: Thanks for the guidance to help us clarify this point. We have changed this CA-rich motif as CCaacAt motif in the text.
Point 7: The name of CATC motif needs to be reconsidered because EMSA and evolutionary conserved sequences suggest CcATC may be more important rather than only CATC.
Response 7: Thanks for the guidance to help us clarify this point. To clarify this, we have changed CATC motif as CcATC motif as suggested.
Point8: In the discussion section, the last and second last sentences (line 272-) seem novel results. Please provide this information in the result section.
Response 8: Thanks for the comment. Even Myb94 and Myb96 from Arabidopsis were shown to bind to the CCaacAt motif, while we did not provide direct evidence of wheat transcription factors binding on this motif. To make clarity, we here have deleted this discussion from the text.
Point9: Line 81, please provide information how two TATA boxes were identified.
Response 9: Following the comment, the description on this point has been added in the text (Line 81-83).
Point10: Line 272, Arabidopsis MYB94 and MYB96 proteins should be written in capital letters.
Response 10: According to the comment, this sentence has been revised.
Point11: In Figure 2 title, remove B. distachyon.
Response 11: According to the comment, “B. distachyon” has been removed in the MS (Line 158).
Point 12: In figure S1, please explain what is SORB 1.
Response 12: We feel sorry for this type error. We have now revised it as “GC(N4)GGCC ” motif in Fig. S1.
Point13: In figure S4a, an error bar is missing.
Response 13: Following the comment, it has been revised (Figure S3a).
We again thank the reviewer for the careful and helpful guidance to improve our study and manuscript.
Reviewer 3 Report
In this study, the authors performed the promoter analysis and EMSA of motifs of TaLTP1 gene, and tried to elucidate cis-elements regulating epidermis specific expression of the gene. They present interesting results, but there are several concerns about this study as below.
1) The authors demonstrate the results of promoter-GUS analyses in transgenic Arabidopsis and B. distachyon as main data. I understand that they employed heterologous system in their experiments since transformation of wheat is quite difficult. However, the data obtained in the heterologous system should be carefully considered. Since the epidermis-specific expression of non-specific LTP genes are conserved in many plant species, promoter analyses of wheat LTP1 gene using Arabidopsis is appropriate. But the authors examined the trichome-specific expression and light response by the same approach. I wonder these expression and response are conserved in Arabidopsis and wheat?
2) The authors performed yeast one-hybrid screening using Arabidopsis TF library and found Myb96 and Myb94 bind to the CA-rich motif in Figure S5. But I regret to say this result has no relevance with the regulation of wheat LTP1 gene. The authors should perform the screening using wheat cDNA library, or at least they need to test binding ability of wheat orthologues of Myb96 and Myb94 to the motif.
3) The authors performed the identification of the cis-elements mainly by promoter deletion analysis, which is a loss-of-function experiment. But gain-of-function experiment is also important for examination of regulatory elements in the promoter. Therefore, the data of Figure S2 should be displayed as an authentic figure instead of supplementary data.
4) It seems that the authors judged GUS staining in the true leaves as an expression of the reporter in epidermis or pavement cells, but is it true? I understand that the –400 bp fragment conferred epidermis-specific expression according to Figure 1c. How about the –380 deletion shown in Figure 2b and –400 to –297 bp fragment in Figure S2?
5) The images of promoter-GUS analyses are so small in Figure 1 and S4. The true leaves are too small to be seen in the panel of –343mCATC in Figure 3d. In addition, I don’t understand what kind of tissue are displayed in Figure 1c since there are no detailed explanation in the figure legend.
6) The details of the mutation of CA-rich and CATC motifs are not explained in Figure 3. The difference in the sequences of wild type promoter and mutated version should be clearly demonstrated as in Figure 2c. I also think the construction of the mutated promoter is not described in Materials and Methods.
7) I think there are some confusion in naming of the motifs. In Abstract, the motif at -380 bp is called ‘GGGCC motif’ (L18), but the authors mention ‘GC(N4)GGCC motif’ in section 2.2. The authors called a motif at –268bp as CATC motif, but its consensus sequence was revealed to be CcATC according to EMSA in Figure 4b, and not CATC but CCATC (or CCATCxATC) is conserved in Figure 4c. If the authors designate the motif by consensus sequence as GC(N4)GGCC motif at –380bp, I think ‘CATC motif’ is not appropriate.
There are also some minor points to be modified or addressed as follows.
8) L20: ‘electrophoretic mobility shift assay (EMSA)’ should be deleted since EMSA can reveal a DNA-protein binding in vitro but cannot reveal the function of a cis-element in vivo.
9) L64: Jeroen et al. is not listed in Reference.
10) L109-114: The result of –400 to –247bp promoter is shown in Figure S2b, which should be mentioned in the text. In addition, ‘35S’ should be changed to ‘CaMV 35S’.
11) L140: I wonder which panel is the results of B. distachyon.
12) L150: No GUS expression is displayed in Figure 3a.
13) L159: ‘the flanking sequence of CA-rich motif’ is vague, which should be clearly mentioned in the text.
14) L160-161: I think ‘supporting CA-motif is exact functional cis-element regulating’ is incorrect since this result does not imply CA-rich motif functions as a cis-element. This phrase should be modified to ‘suggesting this region is not related to the regulation of’.
15) Since the authors used whole cell extract for EMSA (as written in L175), ‘Nuclear protein extract’ in Figure 3a should be changed to ‘Protein extract’.
16) L184: Please indicate the precise region of ‘a short probe harboring a possible CATC repeat motif’.
17) L201-202: Does the phrase ‘the first C nucleotide in CATC repeats is not conserved’ mean that the conserved sequence is C(xATC)(CATC)? If so, I think this phrase is incorrect since the conserved sequence is C(CATC)(xATC) according to Figure 4c.
18) L300: ‘abiotic stress’ is not appropriate to describe the dark treatment in this study.
Author Response
Point 1:
In this study, the authors performed the promoter analysis and EMSA of motifs of TaLTP1 gene, and tried to elucidate cis-elements regulating epidermis specific expression of the gene. They present interesting results, but there are several concerns about this study as below.
1) The authors demonstrate the results of promoter-GUS analyses in transgenic Arabidopsis and B. distachyon as main data. I understand that they employed heterologous system in their experiments since transformation of wheat is quite difficult. However, the data obtained in the heterologous system should be carefully considered. Since the epidermis-specific expression of non-specific LTP genes are conserved in many plant species, promoter analyses of wheat LTP1 gene using Arabidopsis is appropriate. But the authors examined the trichome-specific expression and light response by the same approach. I wonder these expression and response are conserved in Arabidopsis and wheat?
Response 1: Thanks for the guidance addressing this point. As the reviewer mentioned, the epidermis-specific expression of non-specific LTP genes are conserved in many dicot and monocot plants. This includes the conserved expression not only in pavement cells but also in trichomes in aerial tissue.
Even the expression of TaLTP1 had been shown to be responsive to many biotic and abiotic stresses (Genetica, 2010, 138, 843-852), the expression in response to light was shown to be conserved in wheat (revealed by qRT-PCR, Fig. S3a) and in Arobidopsis (Fig. S3b). This might be because of the dual roles of the conserved CATC-motif, which regulates both pavement cell specific expression and light responsive expression. It is noted that the EMSA results indeed showed that the intensity of the shift band changed in response to light when the whole protein extract from light-treated wheat seedlings (Fig. S3c).
Point 2:
2) The authors performed yeast one-hybrid screening using Arabidopsis TF library and found Myb96 and Myb94 bind to the CA-rich motif in Figure S5. But I regret to say this result has no relevance with the regulation of wheat LTP1 gene. The authors should perform the screening using wheat cDNA library, or at least they need to test binding ability of wheat orthologues of Myb96 and Myb94 to the motif.
Response 2: Thanks for the guidance to help us clarify this point. Without the relative evidence in wheat, we have now removed this section for discussion.
Point 3:
3) The authors performed the identification of the cis-elements mainly by promoter deletion analysis, which is a loss-of-function experiment. But gain-of-function experiment is also important for examination of regulatory elements in the promoter. Therefore, the data of Figure S2 should be displayed as an authentic figure instead of supplementary data.
Response 3: Thanks for the helpful comment. We have now reorganized this figure display, with Fig. S2 as a main Fig. 2.
Point 4:
4) It seems that the authors judged GUS staining in the true leaves as an expression of the reporter in epidermis or pavement cells, but is it true? I understand that the –400 bp fragment conferred epidermis-specific expression according to Figure 1c. How about the –380 deletion shown in Figure 2b and –400 to –297 bp fragment in Figure S2?
Response 4: Actually, the promoter constructs for -898, -725, -512, -400 (Fig. 1a), and -380 (Fig. 2b) all showed the expression in pavement cells and trichome in true leaves, which had been validated by histosection anlaysis. The expression in pavement cells did contribute the quantitative activity in leaves, as the expression in trichome is apparently trace. In Fig. 2 (previous Fig. S2), the transgenic Arabiopsis expressing the –400 to –297 bp fragment fused with 35S minimal promoter did not drive the expression in pavement cells lacking the CcATC-motif as shown in Fig. 2. However, the –400 to –247 bp fragment fused with 35S minimal promoter was sufficient to drive the epidermis specific expression including pavement cells and trichomes. To clarify this, we have now added the section results revealing epidermis specific expression in Fig. 2 (Line 129-135).
Point 5:
5) The images of promoter-GUS analyses are so small in Figure 1 and S4. The true leaves are too small to be seen in the panel of –343mCATC in Figure 3d. In addition, I don’t understand what kind of tissue are displayed in Figure 1c since there are no detailed explanation in the figure legend.
Response 5: We feel sorry for the small images in this figure. To show the GUS staining patterns for the whole seedlings (10 days old seedlings shown as the first roll in Fig. 1c and 2 weeks old seedlings shown as the second roll in Fig. 1c), and compare the expression pattern between each construct, we have to put these images in a same panel. For the detail expression in leaf, a zoomed image or leaf histosection were also provided (the third role in Fig. 1c). According to the comment, detail description for this Figure has also been added in figure legend (Line 103-114).
Point 6:
6) The details of the mutation of CA-rich and CATC motifs are not explained in Figure 3. The difference in the sequences of wild type promoter and mutated version should be clearly demonstrated as in Figure 2c. I also think the construction of the mutated promoter is not described in Materials and Methods.
Response 6: According to the comment, we have described the detail for mutation of CA-rich and CcATC motifs in the legend of Fig. 3. Also, the method for achieving mutations were described in Materials and Methods (Line 288-304), and the mutated primers in supplementary Table S1 were described with detailed propose.
7) I think there are some confusion in naming of the motifs. In Abstract, the motif at -380 bp is called ‘GGGCC motif’ (L18), but the authors mention ‘GC(N4)GGCC motif’ in section 2.2. The authors called a motif at –268bp as CATC motif, but its consensus sequence was revealed to be CcATC according to EMSA in Figure 4b, and not CATC but CCATC (or CCATCxATC) is conserved in Figure 4c. If the authors designate the motif by consensus sequence as GC(N4)GGCC motif at –380bp, I think ‘CATC motif’ is not appropriate.
Response 7: We thank the reviewer for helping to clarify this previously imprecise definition.
According to the comment, CATC motif has been revised CcATC motif in the MS, and the name for GC(N4)GGCC motif has been revised in consistence in the manuscript.
There are also some minor points to be modified or addressed as follows.
Point8:
8) L20: electrophoretic mobility shift assay (EMSA) should be deleted since EMSA can reveal a DNA-protein binding in vitro but cannot reveal the function of a cis-element in vivo.
Response 8: According to the comment, the sentence on Line 19-23 has been revised in the MS.
Point9:
9) L64: Jeroen et al. is not listed in Reference.
Response 9: According to the comment, the reference has been added in the MS (Line 63).
Point10:
10) L109-114: The result of –400 to –247bp promoter is shown in Figure S2b, which should be mentioned in the text. In addition, ‘35S’ should be changed to ‘CaMV 35S’.
Response 10: According to the comment, the result for –400 to –247bp promoter fragment
fused with CaMV 35S minimal promoter has been described clearer in the text(Line 129-135). Also, “35S”
in the text has been revised as “CaMV 35S” as suggested.
Point11:
11) L140: I wonder which panel is the results of B. distachyon.
Response 11: Following the comment, we removed “B. distachyon” in the MS (Line 158).
Point12:
12) L150: No GUS expression is displayed in Figure 3a.
Response 12: We feel sorry for this typo. ‘Fig. 3a’ has been revised to ‘Fig. S2a’in the MS (Line 168).
13) L159: ‘the flanking sequence of CA-rich motifis vague, which should be clearly mentioned in the text.
Response 13: Following the comment, we have now revised this sentence with clearer description (Line 178-180).
14) L160-161: I think ‘supporting CA-motif is exact functional cis-element regulating’ is incorrect since this result does not imply CA-rich motif functions as a cis-element. This phrase should be modified to ‘suggesting this region is not related to the regulation of’.
Response 14: Following the comment, we have now revised this sentence with clearer description (Line 180-181).
15) Since the authors used whole cell extract for EMSA (as written in L175), ‘Nuclear protein extract’ in Figure 3a should be changed to ‘Protein extract’.
Response 15: According to the comment, ‘Nuclear protein extract’ in Figure 3a has been revised (Line193).
16) L184: Please indicate the precise region of a short probe harboring a possible CATC repeat motif.
Response 16: According to the comment, the precise region for this probe has been provided in the text (Line198).
17) L201-202: Does the phrase ‘the first C nucleotide in CATC repeats is not conserved’ mean that the conserved sequence is C(xATC)(CATC)? If so, I think this phrase is incorrect since the conserved sequence is C(CATC)(xATC) according to Figure 4c.
Response 17: Thanks for the guidance to help us clarify this point. The consensus of this motif was deduced based on the results both from EMSA and the conserved sequence among nsLTPs. Following the comment, the concesus of CcATC motif has been revised as C(cATC)2, which should more precisely reflect its consensus as revealed by our results (Line 222-224).
18) L300: ‘abiotic stress’ is not appropriate to describe the dark treatment in this study.
Response 18: According to the comment, it has been revised in the MS (Line 322).
We again thank the reviewer for the careful and helpful guidance to improve our study and manuscript.
Reviewer 4 Report
Nice study; however, it needs more details in results section. Figure legends need to be more descriptive and stand-alone. Also get it checked by someone fluent in English.
Line 13: …lipid metabolism et al., while the mechanism.. revise the sentence.
Line 31: These outermost cells….
Line 74: Definition and explanation of TaLTP1 promoter should be required with the previous papers. What do you mean?
Line 86: Previously, we evidenced that….?
Line 125: Full name “electrophoretic mobility shift assay (EMSA” should be mentioned here.
Line 157: ….. was used in EMSA,
Line 182: ..there seems presence of another..
Line 193: … potion?
Lines 193-195: “Moreover, whatever mutation this motif in -400TaLTP1::uidA or -343TaLTP1::uidA constructs did not affect the expression in trichome.” What do you mean?
Line 216: Semi-quantitative RT-PCR revealed that….
Author Response
Response to Reviewer 4 Comments
Point 1: Nice study; however, it needs more details in results section. Figure legends need to be more descriptive and stand-alone. Also get it checked by someone fluent in English.
Response 1: We very appreciate for the encouraging comments about our study. According to the comment, we have revised the manuscript with clearer description in results and figure legend. Moreover, the language has been revised by native English speakers.
Point 2: Line 13: …lipid metabolism et al., while the mechanism.. revise the sentence.
Response 3: Following to the comment, the sentence has been revised in the MS (Line 13).
Point 3: Line 31: These outermost cells….
Response 3: According to the comment, the words have been revised to ‘The epidermis cells…’ in the MS (Line 29).
Point 4: Line 74: Definition and explanation of TaLTP1 promoter should be required with the previous papers. What do you mean?
Response 4: Following the comment, this sentence has now been revised with a clearer description (Line 71-74).
Point 5: Line 86: Previously, we evidenced that….9 ?
Response 4: According to the comment, the word ‘evidenced’ has been revised to ‘demonstrated’ in the MS (Line 89).
Point 6: Line 125: Full name “electrophoretic mobility shift assay (EMSA” should be mentioned here.
Response 6: Following the comment, the full name for EMSA has been added in the MS (Line 142).
Point 7: Line 157: ….. was used in EMSA,
Response 7: Following the comment, it has been revised in the MS (Line 176).
Point 8: Line 182: …..there seems presence of another.
Response 8: Following the comment, it has been revised in the MS (Line 203-204).
Point 9: Line 193: … potion?
Response 9: We feel sorry for this typo. It has been revised as ‘position’ in the MS (Line 214).
Point 10: Lines 193-195: “Moreover, whatever mutation this motif in -400TaLTP1::uidA or -343TaLTP1::uidA constructs did not affect the expression in trichome.” What do you mean?
Response: According to the comment, this sentence has been rewritten to make clarify (Line 211-217).
Point 11: Line 216: Semi-quantitative RT-PCR revealed that…
Response 11: Following the comment, it has been revised in the MS (Line 236).
We again thank the reviewer for the careful and helpful guidance to improve our study and manuscript.
Round 2
Reviewer 2 Report
Dear the editor,
I only have a few minor comments on the revised manuscript.
- Line 24, keywords “trichome”
- Line44, “ZIP IV transcriptional factors” should be “homeodomain-leucine zipper IV transcription factors”
- Line 121, I could not understand the meaning of this sentence “It is noted that the GUS activity in coleoptile and petiole of the transgenic A. thaliana was not observed in transgenic B. distachyon (Fig. 1c).” Since Arabidopsis does not have a coleoptile and Brachypodium does not have a petiole.
- Line 270-271, maize kernel and "Arabidopsis" embryo
- Line 320, it is described that “three to five independent transgenic lines were selected for ---”, but Line 334, “At least 10 independent transgenic lines (T2) were immersed ---” . How many numbers of independent transgenic lines were used for GUS assay? In addition, how about the number of B. distachyon transgenic plants? Please clarify these points.
- Figure legends for Fig S1, S2 and S3 are not found in the files.
Author Response
Point 1: I only have a few minor comments on the revised manuscript. Line 24, keywords “trichome”
Response 1: Thanks for the comment. We have corrected this typo (Line 24).
Point 2: Line 44, “ZIP IV transcriptional factors” should be “homeodomain-leucine zipper IV transcription factors”
Response 2: Thanks for the comment. We have now revised “ZIP IV transcriptional factors” to “homeodomain-leucine zipper IV transcription factors” in the MS (Line 44).
Point 3: Line 121, I could not understand the meaning of this sentence “It is noted that the GUS activity in coleoptile and petiole of the transgenic A. thaliana was not observed in transgenic B. distachyon (Fig. 1c).” Since Arabidopsis does not have a coleoptile and Brachypodium does not have a petiole.
Response 3: We apologize for making the confusion. The ‘coleoptile’ is a typing error, which should be ‘cotyledon’. According to the comment, this sentence has now been revised (Line 122-125).
Point 4: Line 270-271, maize kernel and "Arabidopsis" embryo
Response 4: Thanks for the comment. We have now revised it in the MS (Line 272).
Point 5: Line 320, it is described that “three to five independent transgenic lines were selected for ---”, but Line 334, “At least 10 independent transgenic lines (T2) were immersed ---” . How many numbers of independent transgenic lines were used for GUS assay? In addition, how about the number of B. distachyon transgenic plants? Please clarify these points.
Response 4: Thanks for the comment. To make clarity, we have clearly described this issue in the text (Line 321-326).
Point 6: Figure legends for Fig S1, S2 and S3 are not found in the files.
Response 6: Thanks for the comment. Figure legends for Fig S1, S2 and S3 were added in the MS.
Reviewer 3 Report
In the revised version, the authors responded to my comments properly. I found that the revised manuscript was quite improved and is generally appropriate, but I’d like to ask the authors to check following minor points.
Point 9: I found that Ref 24 was added in the revised version, but this literature is not Jeroen et al but Nieuwland et al.
Point 18: I would suggest the title of section 4.3. as “Expression analysis by Real-Time qPCR”.
Additional point:
As for Fig S3, “CATC motif” should be changed to “CcATC motif” according to the main text.
Author Response
Point 1: In the revised version, the authors responded to my comments properly. I found that the revised manuscript was quite improved and is generally appropriate, but I’d like to ask the authors to check following minor points.
I found that Ref 24 was added in the revised version, but this literature is not Jeroen et al but Nieuwland et al.
Response 1: Following the comment, the literature has been revised in the MS (Line 63).
Point 2: I would suggest the title of section 4.3. as “Expression analysis by Real-Time qPCR”.
Response 2: Thanks for the comment. We have now revised the title of section 4.3. as “Expression analysis by Real-Time qPCR” in the MS (Line 327).
Point 3: As for Fig S3, “CATC motif” should be changed to “CcATC motif” according to the main text.
Response 3: Thanks for the comment. We have now revised “CATC motif” to “CcATC motif” in the Fig S3.
We again thank the reviewer for the careful and helpful guidance to improve our study and manuscript.